# Power shift in the transformation and upgrading of the service sector—Empirical evidence from China

**Xiaochen Geng**🔘*

School of Business Administration, Zhejiang Gongshang University, Hangzhou,Zhejiang Province, China,

* gengxiaochen@ctbu.edu.cn

## Abstract

In recent years, as the global economy has entered the "service economy" period, improving the service industry's total factor productivity, realizing high-quality development, and better playing its supportive role in the real economy and its function of improving people's well-being have become the top priorities for its development. The service industry is growing fast, yet its production and operations are putting pressure on the environment. For this reason, this paper incorporates environmental constraints into the analysis of the transformation and upgrading of the service industry to decompose and analyze the sources of green growth of the service industry. We find that: first, the level of green technology efficiency in the service industry is low, there is a downward trend, and the green technology efficiency gap widens significantly across provinces. Secondly, GTFP in the service sector is declining in general, with large inter-provincial differences, and the obstacle to growth lies in the deterioration of green technical efficiency, while the driving force for growth lies in green technological progress. Thirdly, labor productivity in the service sector has improved considerably, but all at the cost of high inputs, and the growth contribution of green total factor productivity and its constituent factors is negative, and the whole is still in the input factor-driven stage. Finally, the factors affecting GTFP in services are multifaceted, mainly involving structural, environmental regulatory, economic, social and institutional factors. This paper offers a theoretical foundation for further promoting the transformation and upgrading of the service industry to attract jade.

## 1. Introduction and literature review

The scope of China's service industry's development has continued to expand, and its GDP contribution rate has increased year after year, making a remarkable contribution to economic growth. The "14th Five-Year Plan" of China emphasizes the need to accelerate the development of the modern service industry, promote the extension of the productive service industry to specialization and the top of the value chain, and promote the in-depth integration of the modern service industry with the advanced manufacturing industry and modern agriculture. In 2012, the proportion of the tertiary industry in the total economy surpassed the proportion of the secondary industry, and in 2015, it surpassed 50% for the first time. The proportion of the tertiary industry in the total economy surpassed 50% for the first time, and the proportion

**Data availability statement:** All relevant data are within the paper and its Supporting Information files.

**Funding:** We acknowledge the Chinese Ministry of Education Humanities and Social Sciences Research Planning Fund Project (Grant Numbers: No.23YJA790095), awarded to XG, innovative Research Program for Graduate Students of Chongqing Technology and Business University (Grant Numbers: No. CYB240268), awarded to XG. The Chinese Ministry of Education Humanities and Social Sciences Research Planning Fund Project played an important role in study design, data collection, and analysis. For all other funding sources, no funding organization had any role in study design, data collection and analysis, decision to publish, or preparation of the manuscript.

**Competing interests:** The author have declared that no competing interests exist.

of the tertiary industry has been increasing, indicating that China has entered the era of service economy [1], although there is still a large gap compared to the 70% proportion level of developed nations. In addition, the service industry faces several issues, such as development that lags behind economic development, sluggish internal structural upgrading, and low allocation efficiency in various industries [2]. It is evident that China's service industry still has structural flaws, which influence the quality of economic development and the rate of economic transformation to some extent. Based on China's experience, it has been found that the rapid growth of the service sector has come at the cost of high inputs, and that the relative contribution of green total factor productivity is much lower than that of input factor accumulation. GTFP in services is all affected by a variety of factors, most of which are closely related to the supply side. In the new era of development, the transformation and upgrading of the service industry must reconstruct a new development impetus, shifting from the expansion of scale and quantity to the development of quality. High-quality development lies in the modernization of industrial structure, in the transformation of China's growth mode from resource- and capital-based investment-oriented development mode to human capital accumulation and innovation, and, most fundamentally, in innovation-driven, improvement of labor productivity and total factor productivity. Notably, despite the rapid growth of the service sector, its production and operation activities have exerted some environmental pressure. To this end, this paper incorporates environmental constraints into the analysis of transformation and upgrading of the service industry, analyzes the transformation and upgrading of the service industry's power transformation history and future direction from the supply-side perspective with the help of the green economic growth accounting model, and analyzes the influencing factors of the transformation and upgrading of the innovation-driven service industry from the perspective of enhancing the green total factor productivity.

First, the research on the drivers of transformation and upgrading in the service industry focuses primarily on demand and supply drivers. Demand drivers primarily consist of production demand, consumption demand, and urbanization development; the increase in industrial demand for intermediate services has led to the rapid development of productive services [3]; changes in the structure of consumer demand have promoted the upgrading and development of living services, and the increase in urbanization level has led to the increase in consumption demand and public services [3,4]. The primary supply drivers are technological advancement, labor supply adjustments, and industrial policy. Technological innovation can help promote the upgrading and development of the service industry by increasing labor productivity [3,5], and the quantity and quality of labor supply can also help promote the upgrading and development of the service industry to some extent [4]. Technological innovation can help promote the upgrading and development of the service industry by increasing labor productivity [4,5], and the quantity and quality of labor supply can also help promote the upgrading and development of the service industry to some extent [4].

Secondly, the research on the factors influencing the transformation and upgrading of the service industry can be broadly divided into foundational and general factors. Human capital [6,7], technological innovation [8], and the extent of Internet development [9,10] are the most important foundational factors. Human capital directly promotes the transformation and upgrading of the structure of the service industry by increasing the industry's productivity and fostering technological innovation [6,7], increasing R&D investment and accelerating technological innovation can promote advanced structure of China's industry [8], and the level of Internet penetration and the level of Internet development is a key driver for the optimization and upgrading of industrial structure [9,10]. General factors refer to China's macroeconomic development environment, which includes the level of urbanization [2,4], the level of marketization [4,11], the degree of opening up to the outside world [12], government functions

[13,14], population aging [15,16], and numerous other factors. The increase in urbanization and marketization levels can promote the development of the service industry to a certain extent [4], the aging population promotes the development of the production line service industry, which expedites the structural optimization and upgrading of the service industry [15,16], and the government function also influences the transformation and upgrading of the structure of the service industry.

Thirdly, this report examines the evolution of the service industry's structure. China's service industry is lagging in terms of its spatial structure [17,18]. The trend of the evolution of the structure of China's service industry is very similar to that of developed countries, with the proportion of living services, such as catering, declining while productive services, represented by the financial industry, are gradually growing and gaining a higher and higher position in the service economy.

In summary, the results of research on the transformation and upgrading of the service industry's power conversion from the perspective of supply-side structural reform are scarce according to the existing literature. Our research fills a gap in this area. For this reason, this paper incorporates environmental constraints into the analysis of the transformation and upgrading of the service industry, analyzes it from the supply-side perspective, analyzes its future direction at the stage of high-quality development with the help of the green economic growth accounting model, and analyzes its influencing factors.

## 2. Components of economic development's dynamics

Economic growth refers to the growth of social wealth, the growth of production, or the growth of output; economic development refers to the multifaceted socio-economic changes that occur with economic growth, which involve changes in input structure, output, general living standards, and distributional conditions, health and wellness, cultural and educational conditions, the natural environment, and ecology, among other things; economic growth is the motivation and the means of economic development, and economic development is the economic growth's result and purpose [19]. The dynamics of economic development refers to all the motives that drive economic development, and its direct motive and means is economic growth. Therefore, this paper approximates the dynamics of economic growth as the dynamics of economic development, which includes both demand-side and supply-side categories of development dynamics.

### 2.1. Demand-side dynamics based on national accounts

From the demand side, the driving forces of economic development include consumption, investment, and net exports, i.e., the "troika" that drives economic growth. According to Keynesian economic theory, aggregate demand in the economy determines aggregate supply and a state in which the two are equal is the macroeconomic equilibrium state, which determines the size and structure of macroeconomic output. However, in reality, aggregate demand and aggregate supply may not always be equal. It can be seen that in economic development, changes in aggregate demand in the economy play a key role in determining the level of market clearance and the level of development of the economy, i.e., the fundamental driving force of economic development lies on the demand side. The principles of national accounting further indicate that the expenditure approach GDP is equal to the sum of consumption, investment and net exports in the economy. Of these, consumption and investment are derived from the domestic demand for domestic final goods and services by domestic residents (consumers) and firms (producers), respectively, while net exports are derived from the net demand for domestic final goods and services in the international market. Thus, in

economic development, either stimulating domestic demand for consumption or investment or expanding net export demand in external economic development will directly affect the size (including the structure) of gross domestic output, which in turn will drive economic growth and economic development.

## 2.2. Supply-side dynamics based on economic growth accounting

From the supply side, the dynamics of economic development are mainly related to various aspects such as the level of economic development, institutional innovation, government intervention, structural transformation, technological progress and factor endowments. The neoclassical economic growth theory or production frontier theory suggests that total output in an economy depends on changes in the number of input factors and changes in production efficiency. Among them, the quantitative accumulation of input factors is the basic driving force for economic growth, and these factors mainly include traditional factors of production such as labor, physical capital, and land. These factors constitute the material inputs to economic activity, and they lead to economic growth even if there is no qualitative improvement, but merely an increase in quantity. If economic growth due to input factor accumulation stems from changes in the quantity of factors of production, changes in the productivity of input factors are more related to the quality of factors of production and their allocation. In economic academia, economic growth due to changes in input factor productivity is usually attributed to the contribution of total factor productivity (TFP), which portrays all sources of economic growth that cannot be captured by changes in the number of input factors, i.e., the "Solow residual". According to the frontier theory of production, changes in total factor productivity result from technological progress and changes in technical efficiency, which can be further decomposed into changes in pure technical efficiency, changes in scale efficiency, changes in allocative efficiency, and so on.

Relevant theoretical and empirical studies have consistently shown that, under resource and environmental constraints, GTFP progress is the fundamental driving force for long-term sustainable economic growth, and its influencing factors are multifaceted, mainly involving macroeconomics (including factor endowment, economic development, technology diffusion, and industrial structure), economic regulation, environmental regulation, and institutional quality [20]. Among them, environmental regulation and institutional quality are both related to institutional factors; changes in factor endowment (mainly including the composition of labor, land, physical capital, human capital, and intellectual capital) and industrial structure fall under the scope of economic structural transformation; technology diffusion is an important means of promoting technological progress; economic regulation is the main means of government intervention; and the level of economic development will influence the latecomer advantage in the role of GTFP and economic development. Therefore, from the supply side, the driving forces of economic development mainly include the level of economic development, government intervention, institutional innovation, structural transformation, technological progress and factor endowment.

## 3. Growth accounting and analytical ideas for transforming and upgrading the dynamics of the services sector

The dynamics of economic development mainly include supply-side dynamics and demand-side dynamics, the former of which can be analyzed with the help of economic growth accounting models. In this regard, to analyze the transformation of the dynamics of economic development under the dual constraints of resources and the environment, the relative contribution and evolution of factor inputs and total factor productivity changes (including their

constituent factors) in economic development can be dissected from the perspective of green economic growth accounting, and the evolution of the dynamics of economic development and the future direction can be analyzed accordingly.

From the green economic growth accounting model, it can be seen that economic growth can be decomposed into the product of the two components of input factor accumulation (KHC) and total factor productivity change (TFPC), and TFPC can be further decomposed into the product of technical progress (TC) and technical efficiency change (EC), and technical efficiency change can be further decomposed into the product of pure technical efficiency change (PEC) and scale efficiency (PEC) and scale efficiency change (SEC) products of performance. Therefore, to explore the power transformation of economic development, we can analyze the power transformation direction of economic development according to this green economic growth accounting model. To analyze this more intuitively, the natural logarithm of equations ($\underline{1}$) is varied, and the results are shown in equation ($\underline{1}$).

$$\begin{aligned}
ln\left(y\_c\left(k\_c\right)\right)/\left(y\_b\left(k\_b\right)\right) &= ln\left(PEC \times SEC \times TC \times KC \times HC\right) \\
&= pec + sec + tc + kc + hc \\
&= ec + tc + khc \\
&= tfpc + khc
\end{aligned} \tag{1}$$

In equation ($\underline{1}$), ln denotes the natural logarithm of the relevant variable; starting from the second line, the summation terms represent the values of the variables after taking the natural logarithm. For example, tfpc and khc represent the natural logarithm of the change in GTFP and the natural logarithm of capital deepening, respectively, while the left side of the equation represents the natural logarithm of the change in labor productivity.

To measure the relative contribution of GTFP and its components, capital deepening, to economic growth, it is necessary to find the relative contribution of each of the terms in equation ($\underline{1}$) to the change in labor productivity (the left term of the equation). For simplicity, it is straightforward to calculate the share of each of these terms (taking the natural logarithm of the value of each term after the natural logarithm) in the natural logarithm of the change in labor productivity. For example, the relative contribution $tfpc/ln\left(y\_c\left(k\_c\right)\right)/\left(y\_b\left(k\_b\right)\right)$ of GTFP change in economic growth can be derived from, and so on. Based on the evolution of the relative contribution of each growth factor in economic growth, it is possible not only to characterize the evolution of the sources of economic growth but also to analyze the direction of transformation and potential for improvement of the future economic development dynamics in conjunction with the general law of the evolution of economic development dynamics.

## 4. Analysis of the spatiotemporal evolution of GTFP in the service sector

### 4.1. Variable selection

To explore the GTFP of the service industry and its sources of growth, a set of economic value-added and environmental pressure indicators related to economic activities are needed. This study integrates the availability of data and the reasonableness of the selection of indicators, taking the gross output value of the service industry as the desired output, the chemical oxygen demand (COD) in wastewater, the domestic sulfur dioxide (SO2) emissions, and the carbon dioxide (CO2) emissions as the non-desired outputs, and the labor force and the physical capital as the input factors. The selection method of each variable and the data sources are specified below:

**4.1.1. "Desired" output.** Referring to Liu and Zhang (2010) [21]、 Wang and Hu(2012) [22]、Wang et al. (2015) [15], we use the value added of the service sector in each province to express the desired output of the service sector and choose 2003 as the base period for deflating, with the relevant data from the China Statistical Yearbook of past years.

**4.1.2. "Undesired" outputs.** According to Chung et al. (1997) [23], Chen (2010) [24], and Wang et al. (2015) [15], pollution emissions are regarded as the non-expected outputs of service sector production activities. Considering the difficulty of obtaining data on pollution emissions from the service industry in each province in China, based on the methodology of Pang and Deng (2014) [25], chemical oxygen demand (COD), domestic sulfur dioxide (SO2), and carbon dioxide (CO2) are taken as the sources of non-desired outputs from the service industry in this study, and the number of persons employed in the service industry as a percentage of the year-end resident population of each region is multiplied by the current year's domestic The total amount of pollutants discharged in each province is approximated to obtain the pollutant emissions from the service industry in each province. The data on domestic sulfur dioxide and chemical oxygen demand come from the China Environmental Statistics Yearbook, and the data on the number of employed persons in the service industry and the resident population at the end of the year in each region come from the China Statistical Yearbook.

**4.1.3. Capital inputs.** Capital inputs are estimated using the perpetual inventory method to obtain the capital stock, as shown in equation (2).

$$K_t = I_t + (1 - \delta_t)^* K_{t-1} \tag{2}$$

In Equation (2), $K_t$ and $K_{t-1}$ denote the capital stock of region $i$ in period $t$ and $t$-$1$, respectively; $I_t$ denotes fixed asset investment at a constant price of province $i$ in period $t$; and $\delta_t$ denotes the capital depreciation rate of province $i$ in period $t$. According to Wang and Hu (2012) [22]、 Wang et al. (2015) [15], this paper applies the steady state approach proposed by Harberger (1988) [26] to the calculation of the capital stock in the base period, which is shown in equation (3).

$$K_{i,t-1} = I_{i,t} / \left( g_{i,t} + \delta_{i,t} \right) \tag{3}$$

Harberger (1988) [26] pointed out that to reduce the errors caused by short-term output fluctuations and economic cycle fluctuations on the measurement results, the average growth rate over a period of time should be used to express $g_{i,t}$, and this study utilizes the value-added annual average growth rate of the service industry from 2003–2019 to express. According to the research results of Lee and Hong (2012) [27]、 Wang and Hu(2012) [22] and Wang et al. (2015) [15], the depreciation rate of the service industry $\delta_{i,t}$ is uniformly set at 4% in all regions of China. In the calculation, 2003 is chosen as the base period and deflated by the fixed asset index, and the data are obtained from the statistical yearbooks of each province in previous years.

**4.1.4. Labor inputs.** Theoretically, labor input should be considered comprehensively from both quantitative and qualitative aspects. Considering the availability of data, this study uses the product of year-end employment in the tertiary sector and years of education per capita to measure labor input in the service sector. The data come from the China Tertiary Industry Statistical Yearbook, the China Statistical Yearbook and the statistical yearbooks of provinces in previous years.

## 4.2. Data description

After confirming that there is a threshold effect of industrial robot utilization on economic growth, we further discuss if this threshold effect determines whether an economy crosses the

middle-income trap. Did economies in the crossed group leap because the industrial robot intensity had crossed the threshold, while those in the trapped group failed to leap because the threshold was never crossed? We supplement Hypothesis 1 by examining whether there is heterogeneity across groups (trapped group and crossed group) and heterogeneity across time (trapped group before and after falling into the trap, and crossed group before and after crossing the trap) in the contribution of industrial robot applications to economic development using a fixed effects model and a Fisher combination test.

**4.2.1. Heterogeneity analysis of different groups.** In this paper, 30 provinces in mainland China are selected as the sample for analysis, and the general statistical descriptions of the relevant data are shown in Table 1. from which it can be seen that, in general, the development of China's service industry varies greatly over the sample period. From the perspective of inputs, the ratio of the maximum minimum value of capital input elements exceeds 234.58, and the ratio of the maximum minimum value of labor inputs is close to 57.41. From the perspective of outputs, the coefficient of variation coefficients of the non-expected outputs of living sulfur dioxide ($SO_2$) and chemical oxygen demand (COD) in wastewater are relatively close to each other, respectively, 1.125 and 1.168, and the coefficient of variation of carbon dioxide emissions ($CO_2$) has the smallest coefficient of variation of 0.829, which indicates that the development of the service industry is not only in the total amount and the growth rate of large differences but also their environmental pollution pressure is also very large. TFP should be considered in the measurement of energy input and pollution emissions and other non-desired outputs, otherwise, it will lead to bias in the results of the analysis.

## 4.3. Production frontiers and technical efficiency

The input and output data in the development of the service industry in 30 provinces in mainland China from 2003 to 2019 are used as a sample to obtain the green technology efficiency of each province and the composition of the production frontier in the corresponding years, as shown in Appendix A Robustness test of the effect of industrial robots on economic growth in different groups.

First, green technology inefficiency exists widely in the development process of the service industry. From 2003 to 2019, the green technology efficiency of the national service industry was generally low, and the only provinces that had been on the production frontier were Beijing, Tianjin, Shanghai, Jiangsu, Zhejiang, Fujian, Shandong, Henan, Hunan, Guangdong, and Hainan, of which only Beijing and Shanghai had been on the production frontier.

Second, the regional distribution pattern of green technology efficiency in China's service industry did not change significantly during the analysis period, and the inter-provincial differences widened in fluctuations. The ratio of the maximum and minimum values of green technology efficiency in the national service industry increased from 1.8389 to 2.3574, and the

**Table 1. Statistical description of services input, output indicators: 2003–2019.**

| variant | observed value | average value | upper quartile | standard deviation | minimum value | maximum values |
|---|---|---|---|---|---|---|
| gdp | 510 | 4,696.94 | 3,220.56 | 4,583.55 | 137.84 | 25,578.1 |
| k | 510 | 26,103.53 | 16,710.85 | 26,739.68 | 611.14 | 143,359.78 |
| H | 510 | 7,941.96 | 6,971.37 | 5,115.62 | 567.23 | 32,567.41 |
| S02 | 510 | 1.48 | 1.12 | 1.35 | 0.00 | 7.60 |
| COD | 510 | 9.76 | 7.16 | 8.36 | 0.48 | 42.84 |
| CO2 | 510 | 8,568.94 | 6,549.12 | 7,107.23 | 196.24 | 37,957.45 |

coefficient of variation increased from 0.2046 to 0.2280, with a peak value of 0.2261 (2013). It shows that there is a large inter-provincial difference in green technology efficiency in the service sector, and it also shows that the volatility of this difference has increased during the period analyzed. In terms of the dynamics of the regional distribution of green technology efficiency across provinces, provinces with initially lower (higher) green technology efficiency ended up at a relatively lower (higher) level, with most of the correlation coefficients in adjacent years around 0.90, with the upper reaches of the Yangtze River also remaining at a relatively low position.

Third, an analysis of the article's measurement data reveals that there is no inevitable correlation between green technology efficiency in the service sector and the level of service sector development. For example, Hainan, which is relatively backward in terms of the level of service industry development, ranked 28th in terms of value added to the service industry during the sample period, but its green development of the service industry is located on the production frontier in most of the years; whereas Sichuan and Hebei, which have a relatively high level of service industry development, have a green technological efficiency of the service industry that is lower than the national average; for all the provinces of the country, the correlation coefficient between the value added of the service industry and the green technological efficiency of the service industry is not high. The correlation coefficients between the value added of the service industry and the green technology efficiency of the service industry are not high.

## 4.4. GTFP in services and its components

The measurement results of the GTFP of the service industry and its constituent factors for each province in China during the period 2003–2019 are shown in Appendix B and Appendix C. Among them, Appendix B shows the annual decomposition results, and due to space constraints, it only lists the measurement results for the three representative years of 2003–2004, 2011–2012, and 2018–2019; and Appendix C is the cumulative change over the entire period of analysis decomposition results.

First, the GTFP of the service industry has widespread in the development of services. From 2003 to 2019, the cumulative GTFP change and the coefficient of variation across the country were 1.0136 and 0.2471, respectively. With respect to specific provinces, except for Yunnan, which experienced a decrease in GTFP, the GTFP of the other three provinces increased, with Chongqing, Sichuan, and Guizhou increasing by 17.14%, 6.51%, and 1.24% respectively; the highest increase in GTFP in the national service sector was in Beijing, with an increase of 62.80%, followed by Shanghai, with an increase of 54.93%. In terms of specific years, the annual change of GTFP in the service industry is less than 1 in some years, such as 2003–2004, 2011–2012, etc., which indicates that there is a large inter-annual difference in the change of GTFP in the service industry within the analyzed period and that the coefficient of variation creases from 0.0045 volatility to 0.0098, which implies that the inter-provincial gap widens in fluctuation.

Second, the main source of GTFP growth in the service sector lies in green technical progress, while the source of deterioration lies in the deterioration of green technical efficiency. In 2003–2019, the cumulative changes of green technical progress and green technical efficiency in the service industry are 1.1427 and 0.8894, respectively, and they play a significant role in promoting and inhibiting GTFP changes, respectively. The situation in the provinces is also consistent with this, in which the largest green technological progress in the service sector is in Yunnan, with an increase of 13.73%; followed by Chongqing, with an increase of 12.93%; and Guizhou and Sichuan, with increases of 9.80% and 5.95%, respectively. However, there

is a clear polarization in the change of green technology efficiency, in which both Chongqing and Sichuan experienced an improvement in green technology efficiency, with an increase of 3.73% and 0.53% respectively, while Guizhou and Yunnan experienced a deterioration in green technology efficiency, with a decrease of 0.78% and 18.21% respectively. In terms of the whole country, the top three provinces in terms of the increase in green technology progress are Beijing (62.80%), Tianjin (61.16%), and Shanghai (54.93%); green technology efficiency only improved in Guangxi, Jiangsu, Chongqing, and Sichuan, with increases of 51.23%, 32.79%, 3.73% and 0.53% respectively, and Beijing and Shanghai did not undergo relative change.

Thirdly, the deterioration of green technical efficiency is the result of the combined effect of green pure technical efficiency and green scale efficiency, which play a small role but have obvious inter-provincial differences. 2003–2019, the cumulative value of the change in green pure technical efficiency and the change in green scale efficiency is 0.9279 and 0.9636, respectively, which is a decrease of 7.21% and 3.64%, respectively. There is a clear polarization in terms of specific provinces, and the provinces with improved green pure technical efficiency include Chongqing and Sichuan, which have improved by 12.94% and 1.12%, while Guizhou and Yunnan have decreased by 4.09% and 18.44%, respectively; and the green scale efficiency has improved in Yunnan, with an increase of 0.28%, while Chongqing, Sichuan, and Guizhou have decreased by 8.16%, 0.581%, and 3.87%.

## 5. GTFP Growth contribution and potential for enhancement in the services sector

### 5.1. Growth contribution analysis

According to the previous multiple decomposition model of green economic growth, the multiple decomposition of labor productivity changes in the service sector during 2003–2019 is carried out to obtain the growth contribution of green pure technical efficiency changes, green scale efficiency changes, green technical efficiency changes, green technological progress, GTFP changes, and input factor accumulation within the analyzed period, and the results are shown in Appendix D in S1 File.

First, labor productivity has improved, and the interprovincial differences are high. From 2003 to 2019, the national average of labor productivity change mean and its coefficient of variation are 2.7540 and 0.2634, respectively, with large interprovincial differences.

Secondly, the rough-type characteristics of the growth of the service industry are very obvious, and it is still in the input factor-driven stage. From 2003 to 2019, the relative contribution of GTFP in the growth of labor productivity in the service industry was -2.18% and the relative contribution of input factors was as high as 102.18%, with the former negative and the latter positive, and the latter close to 50 times of the former, with the characteristics of sloppy growth being very obvious, and typically in the stage of input-factor drive, with GTFP also deteriorating. Among them, only Beijing and Shanghai's relative contribution of GTFP of the service industry exceeds 50% and is in the innovation-driven stage, respectively 93.67% and 61.46%, while all other provinces are in the input-factor-driven stage, and the contribution of GTFP of 18 provinces is less than 0. It can be seen that the service industry must accelerate the conversion of the development momentum, and realize the transformation of the input-factor-driven to the innovation-driven.

Third, green technological progress in the service industry promotes the growth of labor productivity in the service industry, while green technological efficiency has a dampening effect in it in general. During the analysis period, green technological progress in the service industry promotes the growth of labor productivity in the service industry, with a relative

contribution of 16.31%; changes in green technical efficiency inhibit the growth of labor productivity in the service industry, with a relative contribution of -18.49%, in which the green pure technical efficiency and the green scale efficiency inhibit the growth of labor productivity. These results indicate that green technological progress is the main reason leading to GTFP growth in the service industry and that green technical efficiency and its constituent factors, in general, have hindered labor productivity growth in the analysis period.

Fourth, there are large interprovincial differences in the relative contributions of GTFP and its sources of growth in the development of the services sector. First, there are large interprovincial differences in GTFP in the service industry. Among them, the province with the largest growth contribution is Beijing, whose relative contribution is 93.67%, and the smallest province is Fujian, whose relative contribution is -59.82%. Second, the inter-provincial differences in green technological progress in the service industry are large. Among them, the province with the largest contribution is Beijing, with a relative contribution of 93.67% to labor productivity, and the provinces with the smallest contribution are Fujian and Hunan, with a contribution of 0. Thirdly, there are large inter-provincial differences in the relative contribution of green technology efficiency in the service industry. Among them, Chongqing, Sichuan, Jiangsu, and Guangxi all experienced improvements in green technological efficiency in the service industry and thus promoted labor productivity growth, with relative contributions of 3.15%, 0.43%, 21.48%, and 30.61%, respectively, while the relative contributions of the rest of the provinces were all negative. Fourth, the changes in the green technology efficiency component factors have large inter-provincial differences. Among them, Zhejiang, Jilin, Guangxi, Chongqing, and Sichuan all experienced an improvement in green pure technical efficiency, while all other provinces worsened; Yunnan, Ningxia, Guangxi, and Jiangsu experienced an improvement in green scale efficiency, while all other provinces experienced a deterioration in green scale efficiency. These results also suggest that, although green technological progress is an important way of transforming and upgrading the service industry, improving green technological efficiency also plays an important role, they are both important ways to promote the service industry from factor-driven to innovation-driven.

## 5.2. Enhancement potential analysis

The results of the previous analysis of economic growth accounting show that the contribution of GTFP in the growth of the service industry has been very low since 2003, and the development of the service industry in all provinces is still in the typical input factor-driven stage. In the new development stage, we must accelerate the promotion of the transformation of the service industry's development drive to efficiency-driven and innovation-driven. In our view, there is much room for a power shift in the transformation and upgrading of the service industry, not only in terms of green technological advances but also in terms of improvements in green technological efficiency.

First, the contribution of GTFP growth in the service industry is much lower than the contribution of input factor growth, and there is much room for improvement in future development. Overall, the contribution of GTFP growth in the service industry to its labor productivity growth was -2.18% in 2003–2019, negatively, much lower than the contribution of input factor accumulation. Among them, the contribution of GTFP growth in the service industry in all provinces is also much smaller than the contribution of input factors, and they all remain in the input factor-driven stage of input typical. This shows that economic growth in the service sector is still typical of crude economic growth, and all provinces have a long way to go in converting to innovation-driven growth, which also means that there is a lot of potential space for them to realize transformation and upgrading through the conversion of development power.

Second, the potential room for improvement in green technological efficiency and green technological progress is large, especially in green technological efficiency, not only in terms of greater room for adjusting and optimizing the scale of production but also in terms of greater potential for more effective use of best-practice technologies. During the period 2003–2019, the average relative contributions of green technology progress and green technology efficiency changes in the growth of the service sector are 16.31% and -−18.49%, respectively, both of which still have large potential for improvement. Among them, the average growth contribution of green technology progress has a huge room for improvement compared with that of Beijing (93.67%), while the improvement of green technology efficiency has a significant gap compared with that of Jiangsu (21.48%). It can be seen that the service industry in most provinces has a large space for green technology progress and green technology efficiency improvement, especially green technology efficiency has a larger space for improvement. In addition, in terms of the source of change in green technical efficiency, the average growth contribution of green pure technical efficiency and green scale efficiency in the service industry in the region during the analysis period was −12.57% and −5.92%, respectively, which suggests that as a whole, the optimal scale of production has not yet been realized in the development of the service industry, nor has the best-practice technology been used to the fullest extent. Therefore, in future development, by adjusting the production scale of their service industry and more effectively using the best practice technology in the economy, both can promote the efficiency of their service industry, and then promote the change of their development drive to innovation-driven.

## 6. Analysis of factors influencing GTFP in the service sector

Productivity measures are reliable only if they correctly take into account energy and environmental factors in the framework of sustainable development, otherwise, productivity may be overestimated or underestimated [28]. Total factor productivity without considering energy saving and emission reduction cannot truly reflect specific technological and efficiency changes. Previously, under the resource and environmental constraints, green growth accounting was carried out for the service industry, which portrayed the evolution of the dynamics of the service industry's development, and it was found that the fundamental driving force for sustainable economic development in the future lies in the enhancement of the level of GTFP. To promote the GTFP enhancement of the service industry, it is also necessary to clarify its influencing factors to have a target in economic development. To this end, this section empirically analyzes the influencing factors of GTFP in the service industry based on existing relevant studies and combined with the panel data of Chinese provinces.

### 6.1. Synthesis of relevant studies

Improving GTFP in the service industry is the key to the transformation, upgrading, and high-quality development of the service industry, and it is of great practical significance to clarify it's influencing factors. Compared with industry, academic attention to GTFP in the service sector is relatively limited. In recent years, several studies have explored this issue and found that the influencing factors of GTFP in the service industry are multifaceted, mainly including the internal structure of the service industry, the level of economic development, the structure of energy consumption in the service industry, the level of urbanization, the environmental regulations, the degree of informatization, the degree of openness of the economy, the human capital, and factor endowment, etc. These studies mainly reflect the following aspects. These studies are mainly reflected in the following aspects:

First, the measurement and comparative analysis of service industry TFP and GTFP found that they are quite different. Pang et al. (2011) [29] took the lead in focusing their research on China's service industry, and based on the directional distance function and the Malmquist-Luenberger index, they conducted a comparative analysis of China's total factor productivity in the industrial and service industries from 1998 to 2012 and found that the GTFP of the industry was generally lower than that of the service industry in terms of quantity, but the former has a slightly higher growth rate. Subsequently, Wang et al. (2015) [15] used the SBM model and the serial ML index to evaluate the traditional TFP and GTFP of China's service industry from 2000 to 2012 based on inter-provincial and industry panel data respectively and concluded that the differences between the two were very obvious, with the average annual growth rates of traditional TFP and GTFP under the sub-industry perspective being 4.9% and 1.9% respectively, and the corresponding values under the inter-provincial perspective being 4.9% and 1.9% respectively. The average annual growth rates of traditional TFP and GTFP in the sub-industry perspective are 4.9% and 1.9% respectively, while the corresponding values in the inter-provincial perspective are 4.7% and 3.2%.

The second is to empirically analyze the influencing factors of GTFP in the service industry. Pang and Wang (2016) [30] constructed the Bootstrap two-stage analysis method model to measure the GTFP of China's service industry based on the panel data of each region and analyzed the influencing factors of the GTFP of the service industry, and the results found that the green transformation of China's service industry still has a long way to go [15], on the other hand, selected the inter-provincial panel data from 2012 to 2014, constructed the SBM-GML model to measure and decompose the environmental TFP of China's service industry, and analyzed the impact of FDI in the service industry on GTFP. In addition, Chen and Wang (2020) [31] examined the spatial differences in the growth of GTFP in China's service industry and the factors affecting it and found that the level of economic development, the level of the service industry, and the increase in the level of human capital significantly inhibit the growth of GTFP in the service industry, whereas FDI in the service industry, the level of urbanization, and the fiscal expenditures are conducive to the improvement of the GTFP in the service industry. Teng (2020) [32] examined the spatial differentiation of GTFP growth in the service sector and its drivers in China and found that the intensity of energy consumption under the condition of variable returns to scale has the largest individual driving effect on the spatial differentiation of GTFP in the service sector, whereas the level of environmental regulation under the condition of constant returns to scale has the largest individual driving effect on the spatial differentiation of GTFP in the service sector, and that the spatial differentiation of GTFP in the service sector is more a function of each driver than the other. differentiation is more a result of the interaction of the drivers. Peng (2020) [33] examined the impact of heterogeneous environmental regulation on the GTFP of the service industry and found that the impact of formal environmental regulation on the GTFP of the service industry has a U-shaped structure and that the improvement of the level of development of the service industry can improve the GTFP of the service industry, whereas the informal environmental regulation promotes the GTFP of the service industry, but the improvement of the level of development of the service industry will weaken the promotion effect of the informal environmental regulation on the TFP of the service industry. TFP promotion effect of informal environmental regulation.

Third, the constituent factors of GTFP in the service industry are analyzed by influencing factors. Zeng (2019) [34] found that energy structure, industry structure, human capital accumulation, and capital deepening are all conducive to the growth of GTFP in the service sector and that it is mainly through the influence of cumulative green technological progress changes that have an impact on GTFP. Meng et al. (2021) [35] examined the influencing

factors of green technical efficiency in China's service industry and found that the level of economic development has a U-shaped relationship with green technical efficiency in the service industry and that industrial integration, energy structure, and labor quality contribute to the improvement of green technical efficiency in the service industry.

## 6.2. Variables and data selection

**6.2.1. Variable selection.** As can be seen from the measurement of GTFP in services, it is related not only to value added or gross output and input factors that portray desired output but also to pollutants that portray non-desired output. In other words, the factors affecting GTFP relate to the economy, resources, and the environment. The relevant studies reviewed earlier also indicate that factors that play a role in energy and resource use efficiency, productivity development level, output efficiency, and ecological and environmental conditions in the development of the service sector should be included in the analysis. Based on the existing studies, this study categorizes the potential influencing factors of GTFP in the service industry into five broad categories, including structural factors, environmental regulation factors, economic factors, social factors, and institutional factors, as shown in Table 2. In the empirical analysis, combined with the availability of data, the specific measurement indicators for each broad category are screened, and the specific variables are selected as follows:

**Structural factors:** Different service industry structures have different levels of green development and effective degrees of greenness, which in turn have an impact on resource allocation, technological innovation, and GTFP. Different energy structures lead to different environmental pollution control costs and profit margins of enterprises, which in turn affect the GTFP of the service industry. In addition, the basic input factor ratio in the production process of the service industry has a direct impact on the GTFP of the service industry. Therefore, this study categorizes structural factors into three dimensions: internal structure of the service industry (big), factor structure (ysjg), and energy structure (nyjg).

The internal structure of the service sector (nbjg). In terms of China's end-use energy consumption, "transportation, warehousing, and postal transportation", "wholesale and retail trade", and "accommodation, food and beverage" account for more than 70% of the energy consumption in the service sector. More than 70% of the energy consumption is in the service sector. Generally speaking, an increase in the proportion of high-value-added, high-efficiency, and high-knowledge-intensity industries will increase the level of green development of the service industry, while an increase in the proportion of high-energy-consuming service

**Table 2. List of potential influences on GTFP in the service sector.**

| form | variant | Measurement indicators |
|---|---|---|
| Structural factors | industry structure (*nbjg*) | Share of value added of three sectors, namely, accommodation and catering, wholesale and retail trade, and transportation, storage and postal services, in the value added of the service sector |
| | Elementary structure (*ysjg*) | capital-labor ratio |
| | energy structure (*nyjg*) | Share of industrial coal energy consumption in total industrial energy consumption |
| restraint | Administrative environmental regulation (*es*) | Investment in pollution control as a share of GDP |
| | Market-based environmental regulation (*er*) | Revenue from sewage charges as a share of GDP |
| economic factor | Level of development of services (*savl*) | Value added of services as a share of GDP |
| | Level of economic development (*pgdp*) | Natural logarithm of GDP per capita |
| social factor | degree of informatization (*ict*) | Internet penetration |
| Institutional factors | Government regulatory capacity (*govp*) | Fiscal expenditure as a share of GDP |

industries will reduce the degree of green effectiveness of the service industry. For this reason, we choose "transportation, warehousing, and postal services", "wholesale and retail trade" and "accommodation and catering" as the proportion of the value added of the three energy-consuming traditional distribution services in the tertiary industry. The proportion of value added of the three energy-intensive traditional distribution service industries, namely "transportation, storage, and postal services", "wholesale, retail trade" and "accommodation and catering", in the tertiary industry is used as an indicator of the internal structure of the service industry and is denoted by nbjg.

Energy Structure (nyjg): China's coal accounts for the largest share of all energy consumption, and the coal-based energy consumption structure is not conducive to the sustainable development of the economy, and serious pollution of the environment. In recent years, the proportion of energy consumption in China's service sector has gradually increased and the proportion of industrial energy consumption has gradually decreased. Pang and Wang (2016) [30]found that the decline in the proportion of coal consumption in the service sector of the service industry helps to increase the GTFP of China's service industry, which is consistent with the findings of Wang and Wang (2017) [36]. In terms of the drivers of GTFP spatial differentiation in China's service sector, Teng (2020) [32] shows that the intensity of energy consumption under the condition of variable scale compensation has the largest individual driving effect on the spatial differentiation of GTFP in the service sector. For this reason, this study measures the energy consumption structure of the service sector in terms of the share of coal consumption of the service sector in the total energy consumption of the service sector in each province, which is denoted by ES.

Factor structure (ysjg): The capital-labor ratio reflects the factor endowment of the service sector, with a lower ratio indicating a bias towards labor intensity and vice versa for capital intensity. Differences in the relative input shares of capital and labor can have an impact on the GTFP of the service sector, and this paper uses the share of capital stock and employment in the service sector in each province to measure the capital-labor ratio

**Environmental regulatory factors:** GTFP is total factor productivity under resource and environmental constraints, and the state of the environment will undoubtedly have a direct impact on its measurement. Therefore, artificial environmental regulation may also have an impact on environmental conditions, which in turn may also affect GTFP. Existing research also suggests that there may be a "compliance cost effect" and an "innovation effect" of environmental regulation in economic development. Therefore, we also include environmental regulations in our analytical model. As an external constraint on the behavior of the service industry, environmental regulation has a direct impact on the revenue, transaction costs, and management efficiency of the service industry, which in turn affects the GTFP change. For example, Teng (2020) [32] finds that the level of environmental regulation alone drives the largest spatial divergence of GTFP in the service sector, with constant returns to scale.

At present, there is no specific indicator to measure environmental regulation intensity in China, and the relevant metrics used by academics are not uniform. Generally speaking, environmental regulation intensity is a reflection of the cost of pollution control for enterprises, and the greater the environmental regulation intensity, the higher the cost of pollution control for enterprises. Teng (2020) [32] adopt total investment in environmental pollution control and total investment in environmental pollution control as a proportion of GDP as a measure of environmental regulation, respectively. In addition, the amount of sewage fee collection is also an important indicator of environmental regulation, and Pang and Wang(2016) [30] used the proportion of sewage fee revenue to GDP to measure environmental regulation. In this study, the proportion of sewage fee revenue to GDP (er) is selected to measure the intensity of

environmental regulation, and the proportion of total investment in environmental pollution control to GDP (es) is used for the robustness test.

**Economic factor:** Total factor productivity is an important source of economic development, and it is also influenced by the level of economic development. Generally speaking, the higher the level of economic development, the more inputs are available for R&D activities, so that economic activities are no longer overly reliant on natural resources, but more inclined to promote economic growth through technological advances and efficiency improvements. In addition, macroeconomic stability has an impact on the smooth running of R&D activities, which may also affect the changes in GTFP. To this end, we mainly explore the impact of the level of economic development on GTFP in the service industry from two aspects, namely, the level of regional service industry development (savl) and the level of regional economic development (pgdp).

Level of service sector development (sval). A higher share of value added to the service sector means a more developed service economy in the region. However, there are mixed reviews about the impact of the development level of the service industry on the GTFP of the service industry. For example, Wang and Wang (2017) [36] argue that the level of service industry development not only does not have a promotional effect on the increase of GTFP in the service industry, but also has a significant inhibitory effect, and Chen and Wang(2020) [31] and others have reached the same conclusion. To analyze this, we measure the level of service industry development by the ratio of the value added of the service industry to the gross regional product of each province.

Regional economic development level (pgdp). Pang and Wang(2016) [30] calculated the environmental total factor productivity of the service sector in 30 provinces and cities in China during 2010–2013 under the green growth framework and found that the effective degree of the service sector under the resource and environmental constraints showed a positive "U-shaped" relationship with the level of economic development. Chen and Wang (2020) [31] found that an increase in GDP per capita significantly inhibits the growth of GTFP in the service sector. For this reason, we use GDP per capita to measure the level of economic development and apply the GDP per capita index to deflate and take its natural logarithm as a measure. In addition, a quadratic term for GDP per capita is added to the regression as a way to test whether there is a nonlinear effect of the level of economic development on total factor productivity in the service sector.

**Social factor:** The development of infrastructure is more conducive to promoting industrial agglomeration, providing space for scale efficiency, and the agglomeration effect and diffusion effect brought by the infrastructure can drive the dissemination and diffusion of information, technology, talent, and other resources in different regions (industries), thus enhancing the GTFP of the service industry, and at the same time can also make the service industry pollution control costs to be apportioned or the scope of pollution to be further expanded, which affect the GTFP of the service industry. In the context of the digital economy, information infrastructure is crucial. The promotion and popularization of information technology are crucial, which can alleviate information asymmetry, reduce logistics costs and other transaction costs, and stimulate the network economy effect in economic development, which may affect the level and direction of technological progress, and thus affect the level of GTFP in the service industry. For this reason, the Internet penetration rate (ict) is selected to measure the degree of informatization in this study.

**Institutional factors:** Numerous studies at home and abroad have shown that there is a high degree of positive correlation between the quality of institutions and economic development and that good institutions are not only conducive to the smooth running of productive activities but also to the effective allocation of resources. Market failure is one of the unavoidable

realities in economic development, and reasonable government regulation can make up for it. However, the policy effect usually has a time lag, so the impact of government regulation on GTFP is difficult to directly reflect but also may cause distortion. On the one hand, the government can further improve the allocation of social factors of production through fiscal expenditure, tax policy, and industrial policy, and thus promote the growth of GTFP [6–38]. However, if the intensity of the government's macro-control policies is too great, it will lead to its inability to release the economic vitality of enterprises, leading to inefficient resource allocation, and thus hindering the growth of GTFP [39]. Some empirical studies have found that government regulation of the economy can have an important impact on the development of the service industry. For example, Pang and Wang (2016) [30] argues that strengthening government regulation of the economy can enhance the effective degree of the service industry, and Chen and Wang (2020) [31] argue that the expansion of fiscal expenditure contributes to the improvement of GTFP in the service industry. For this reason, this study selects government regulation capacity (govp) as an indicator to measure the system, specifically using local fiscal expenditure as a share of GDP to characterize it.

In addition, green technical progress (GTC) and green technical efficiency change (GEC) are both decomposition factors of GTFP (GTFP). The explanatory variables in this section are the GTFP index (GTFP) of the service industry and its decomposition factors measured in the previous section. Considering that the changes in GTFP of the service industry in some provinces and regions in adjacent years are very small, the cumulative GTFP index, the green technical efficiency change index, and the green technical progress index during the sample period are used as the explanatory variables here, and such a treatment does not change the trend of the changes in these variables, but it will make the trend of the changes easier to recognize. The relationship between the influence of each influencing factor and the GTFP and its components is shown in Fig 1.

**6.2.2. Data description.** The relevant data in this part are mainly based on the relevant data from the publicly available China Statistical Yearbook, China Environmental Statistical Yearbook, China Energy Statistical Yearbook, and China Science and Technology Statistical Yearbook, and some of the data are based on the results of related studies. The relevant data are all inter-provincial panel data, in which the sample space of the data is 2004–2019, and the general statistical descriptions of the relevant variables are shown in Table 3.

To avoid the problem of multicollinearity in the variables employed, we used the VIF variance inflation factor test and the results are shown in Table 4. From the results, the maximum VIF

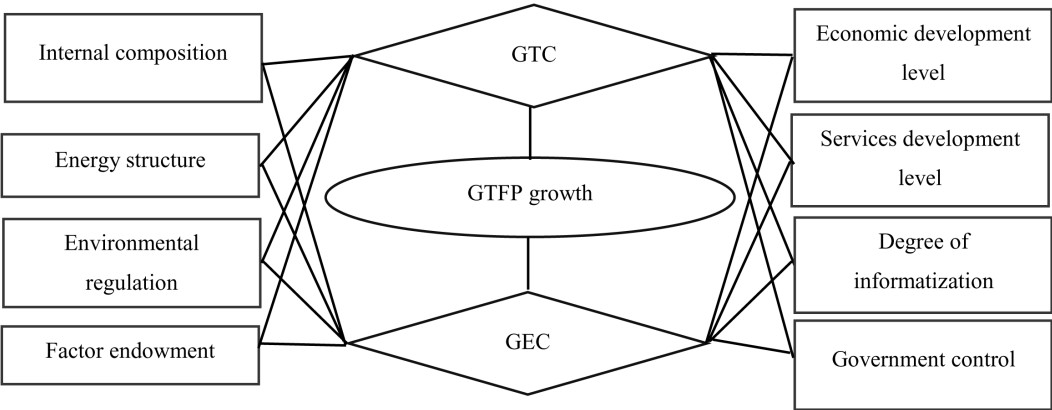

**Fig 1. Factors influencing GTFP in the service sector.**

value is 7.02, which is less than the critical value of 10, indicating that there is no problem of multicollinearity in the variables selected for this study.

## 6.3. Empirical model setting

In the benchmark regression, we first use the most commonly used fixed-effects model and random-effects model in dealing with panel data to conduct the regression, and the specific model is shown in Equation 4.

$$Y_{it} = \beta_0 + \beta_i X_{it} + \gamma_t + \mu_i + \varepsilon_{it} \tag{4}$$

where $Y_{it}$ is the explanatory variable; $X_{it}$ is a series of control variables; $\gamma_t$ represents the time fixed effect, which measures the tendency over time for regions not included in the control variables; $\mu_i$ represents the fixed effect of province, which represents the variable that changes with individuals but not over time; $\varepsilon_{it}$ is the residual term; and the subscripts $i$ and $t$ represent province $i$ and year $t$, respectively. According to the results of the Hausman test (see Table 5), both the GTFP and its constituent factors, their influencing factor analysis models significantly rejected the original hypothesis, so we used the fixed effect model.

## 6.4. Analysis of model estimation results

### 6.4.1. Benchmark regression results.
The results of the benchmark regression are shown in Table 6, from which at least the following conclusions can be drawn:

First, the decline in the share of traditional services contributes to the improvement of green technical efficiency in the service industry, but the impact on green technical

**Table 3. Statistical Description of Factors Influencing GTFP in Service Industries.**

| variant | observed value | average value | upper quartile | standard deviation | minimum value | maximum values |
|---|---|---|---|---|---|---|
| *nbjg* | 480 | 0.45 | 0.46 | 0.09 | 0.21 | 0.77 |
| *lnpgdp* | 480 | 10.09 | 10.10 | 0.67 | 8.29 | 11.86 |
| *es* | 480 | 0.02 | 0.01 | 0.05 | 0.00 | 0.41 |
| *ysjg* | 480 | 5.50 | 4.57 | 3.96 | 0.70 | 26.46 |
| *sval* | 480 | 0.44 | 0.41 | 0.10 | 0.27 | 0.84 |
| *nyjg* | 480 | 0.10 | 0.06 | 0.12 | 0.00 | 0.58 |
| *ict* | 480 | 0.36 | 0.39 | 0.20 | 0.03 | 0.78 |
| *govp* | 480 | 0.22 | 0.20 | 0.10 | 0.08 | 0.63 |
| *hc* | 8.76 | 1.02 | 8.71 | 6.38 | 12.78 | 8.76 |

**Table 4. Results of Multiple Covariance Test of Factors Influencing GTFP in Service Industry.**

| variable name | VIF | 1/VIF |
|---|---|---|
| *ict* | 6.55 | 0.1527 |
| *lnpgdp* | 7.02 | 0.1425 |
| *sval* | 2.17 | 0.4618 |
| *ysjg* | 1.64 | 0.6116 |
| *es* | 1.20 | 0.8332 |
| *nyjg* | 1.20 | 0.8332 |
| *govp* | 1.98 | 0.5062 |
| *nbjg* | 1.60 | 0.6265 |
| Mean VIF | 2.91 | |

progress and GTFP is not significant. Among them, the effect of the internal structure of the service industry on the green technical efficiency of the service industry is significantly negative, which indicates that the improvement of the internal structure of the service industry, i.e., the decline in the share of the traditional high-energy-consuming and high-emission distribution service industry in the service industry significantly improves the green technical efficiency of the service industry. The effect of changes in the internal structure of the service industry on the enhancement of both GTFP and green technological progress is insignificant, with a positive coefficient estimate for green technological progress and a negative coefficient estimate for GTFP. The possible reason behind this is that although the financial and real estate service industries with low energy consumption and low pollution have grown rapidly in recent years, there is a bubble in their development [30], and China's financial industry has deviated from the real economy to a certain extent, and they have largely failed to effectively promote the green technological progress of the service industry and the enhancement of GTFP.

Second, there is a significant U-shaped nonlinear relationship between the level of regional economic development and GTFP in the service industry. Among the model estimation results of GTFP and green technological progress, the coefficients of the quadratic terms of log per capita GDP are all positive, and the coefficients of the primary terms are all negative, which indicates that the relationship between the GTFP of the service industry, the green technological progress and the level of economic development are all non-linear, and that only when the level of economic development reaches a certain degree will it promote the green technological progress of the service industry and GTFP enhancement. The reason may lie in the fact that the development of the service industry, like agriculture and industry, initially starts from low-end labor-intensive industries with lower thresholds, and their technological level is often lower. However, as the level of economic development continues to rise, especially the manufacturing industry's increasing demand for medium- and high-end productive services, this will greatly promote green technological progress in the service industry, which in turn will promote GTFP.

Third, environmental regulations contribute to green technological progress, but their positive impact on GTFP is not significant. In particular, administrative environmental regulation has a significantly positive impact on green technological progress in the service sector, reflecting to some extent that the increase in government investment in pollution control has pushed service sector firms to accelerate the pace of innovation to reduce costs, thus promoting green technological progress. However, as the impact of environmental regulation on green technological efficiency is not significant, this to some extent offsets its positive impact on green technological progress, thus showing a non-significant impact on GTFP. We believe that the reason for this is that administrative environmental regulation is inevitably "one-size-fits-all" in its implementation, which leads to a dilution of the resource optimization effect of environmental regulation or even a large negative impact

**Table 5. Hausmann test results.**

| Test: Ho: difference in coefficients not systematic | | |
| --- | --- | --- |
| chi2(8) = (b-B)'[(V_b-V_B)^(-1)](b-B) | | |
| Tfp | tc | te |
| = 25.68 | = 82.39 | = 46.65 |
| Prob>chi2 = 0.0023 | Prob>chi2 = 0.0000 | Prob>chi2 = 0.0000 |

Note: In the table, tfp, tc and te characterize GTFP, green technical progress and green technical efficiency, respectively.

(the coefficient estimate here is negative but not significant), thus inhibiting the improvement of green technology efficiency and GTFP. This inhibits the improvement of green technology efficiency and GTFP.

Fourth, none of the effects of factor endowment structure on GTFP and its sources of growth in the service sector are significant. Among them, the coefficient estimates of the capital-labor ratio and the GTFP of the service industry and its constituent factors do not pass the significance test, which suggests that the causal relationship between them is not obvious. The possible reasons for this are: on the one hand, in the stage of high economic growth, capital deepening mainly relies on the expansion of crude production scale, which is evident from the previous growth accounting results, which will increase the ecological environment pressure; on the other hand, capital deepening also promotes the gradual conversion of enterprise production from labor-intensive to capital-intensive, and capital-intensive enterprises tend to be more technologically advanced, and the improvement of technological level again To a large extent, it offsets its negative impact on ecological resources and the natural environment. Therefore, under the action of these

**Table 6. Estimation results of factors affecting GTFP and its components in the service sector.**

| variable | Green total factor productivity | Green technological advances | Green technology efficiency |
|---|---|---|---|
| nbjg | −0.115 | 0.104 | −0.284*** |
|  | (−1.51) | (1.37) | (−3.33) |
| lnpgdp | −1.502*** | −1.593*** | −0.011 |
|  | (−8.46) | (−9.05) | (−0.05) |
| lnpgdp² | 0.065*** | 0.073*** | −0.003 |
|  | (6.64) | (7.54) | (−0.26) |
| es | 0.294 | 0.407** | −0.052 |
|  | (1.47) | (2.05) | (−0.23) |
| ysjg | 0.002 | −0.002 | 0.004 |
|  | (0.78) | (−1.06) | (1.50) |
| sval | −0.265*** | −0.174* | −0.045 |
|  | (−2.82) | (−1.87) | (−0.43) |
| nyjg | 0.056 | 0.142*** | −0.068 |
|  | (1.15) | (2.96) | (−1.26) |
| ict | −0.664*** | −0.638*** | −0.105 |
|  | (−6.20) | (−6.00) | (−0.88) |
| govp | −0.340*** | 0.086 | −0.340** |
|  | (−2.62) | (0.67) | (−2.36) |
| _cons | 9.999*** | 9.374*** | 2.187** |
|  | (10.79) | (10.22) | (2.16) |
| Year FE | Yes | Yes | Yes |
| Province FE | Yes | Yes | Yes |
| N | 480 | 480 | 480 |
| F | 14.9827 | 17.7058 | 4.8223 |
| R² | 0.817 | 0.823 | 0.747 |

Note:

*,

**, and

***indicate significance levels of 10%, 5%, and 1%, respectively; numbers in parentheses are standard errors.

two opposing forces, the impact of capital deepening on GTFP and its components is ambiguous.

Fifth, the impact of the level of service industry development on its GTFP is significantly negative. Among them, the coefficient estimates of the level of development of the service industry in each of the 3 models are negative, and only the green technical efficiency model is not significant, which indicates that the improvement of the level of development of the service industry does not promote the improvement of its GTFP. The reason for this is that we believe it is related to the crude growth model of China's service industry in recent years. That is to say, in the input factor-driven stage (the previous growth accounting results show that the growth contribution of GTFP is negative), the development of China's service industry is basically to increase the number of factor inputs to obtain rapid growth, which ignores the role of technological progress and does not pay great attention to energy saving and emission reduction, which results in the inhibition of the level of GTFP and its constituent factors. This is consistent with the findings of Wang and Wang (2017) [36], who found that the level of development of the service industry not only does not have a promotional effect on the GTFP of the service industry, but also produces a significant inhibitory effect, and Chen and Wang(2020) [31]. also came to the same conclusion.

Sixth, the effect of energy results on GTFP in the service industry is not significant. Among them, the effect of energy consumption structure on green technological progress is significantly positive, indicating that energy consumption structure has a facilitating effect on green technological progress in the service industry. However, its relationship with both green technical efficiency and GTFP is not significant, and from the sign of the coefficient estimates, it has an inhibitory effect on the improvement of green technical efficiency. Related studies have also found that the coal-dominated energy consumption structure has an inhibitory effect on GTFP in the service industry [30–36].

Seventh, the degree of informatization has a significant negative impact on both GTFP and green technological progress in the service sector, with a non-significant inhibitory effect on green technological efficiency. This may be explained by the fact that, while IT technology is applied on a large scale, to give full play to its networked technological advantages and improve operational efficiency requires matching management innovation, model innovation and institutional innovation, as well as human capital enhancement, business restructuring and organizational change and other changes in the institutional mechanism that are compatible with it, which requires that China's service industry development break through the traditional model and transform to a modern service industry development model, or else it will Foster the development of some low-end service industries, thus worsening the GTFP. In this regard, China's service industry in general has done far enough, and there is more room for improvement [30–36].

Eighth, government intervention inhibits GTFP and green technical efficiency improvement in the service industry, and it has no significant effect on green technological progress. Among the estimation results of the GTFP and green technical efficiency models, the intensity of government intervention (GDP share of fiscal expenditure) is significant at the 1% significance level, which indicates that government intervention inhibits GTFP and green technical efficiency improvement in the service industry. The government's intervention in economic activities through fiscal subsidies and tax incentives-based fiscal expenditure policies compensates for market inefficiencies to a certain extent, but the artificial redistribution of production resources also disrupts the market price mechanism and the competition mechanism, and they cause efficiency losses, which deteriorate the service industry's GTFP and green technical efficiency.

## 6.5.  robustness check

To further verify the reliability of the empirical results, the following samples are analyzed for robustness using two commonly used methods, namely, the method of considering replacement variables and the method of supplemental variables, respectively, and the results are shown in Table 7. Among them, models (1)-(3) remeasure GTFP by adjusting the undesired output variables, whereby the explanatory variables are replaced; models (4)-(6) replace the environmental regulation variables with sewage charges revenue as a share of GDP instead; and models (7)-(9) introduce the level of human capital (*hc*). The estimation results of

**Table 7. Robustness tests of GTFP Influencing Factors in the Service Sector.**

| variable | (1) Green total factor productivity | (2) Green technological advances | (3) Green technology efficiency | (4) Green total factor productivity | (5) Green technological advances | (6) Green technology efficiency | (7) Green total factor productivity | (8) Green technological advances | (9) Green technology efficiency |
|---|---|---|---|---|---|---|---|---|---|
| *nbjg* | −0.061 | 0.224*** | −0.345*** | −0.114 | 0.116 | −0.294*** | −0.046 | 0.211*** | −0.313*** |
| | (−0.85) | (3.18) | (−3.51) | (−1.51) | (1.54) | (−3.48) | (−0.64) | (2.99) | (−3.19) |
| *lnpgdp* | −1.462*** | −1.103*** | −0.314 | −1.264*** | −1.503*** | 0.169 | −1.466*** | −1.099*** | −0.323 |
| | (−8.75) | (−6.75) | (−1.37) | (−6.54) | (−7.76) | (0.78) | (−8.81) | (−6.74) | (−1.43) |
| *lnpgdp²* | 0.063*** | 0.059*** | −0.002 | 0.051*** | 0.068*** | −0.013 | 0.063*** | 0.058*** | −0.001 |
| | (6.82) | (6.53) | (−0.13) | (4.80) | (6.35) | (−1.10) | (6.85) | (6.53) | (−0.11) |
| *es* | 0.385** | 0.343* | 0.058 | | | | 0.379** | 0.349* | 0.044 |
| | (2.05) | (1.86) | (0.22) | | | | (2.02) | (1.90) | (0.17) |
| *er* | | | | 39.900*** | 11.582 | 33.251** | | | |
| | | | | (2.75) | (0.80) | (2.05) | | | |
| *ysjg* | 0.001 | −0.005** | 0.007** | 0.002 | −0.002 | 0.003 | 0.001 | −0.005** | 0.007** |
| | (0.63) | (−2.42) | (2.32) | (0.70) | (−0.95) | (1.33) | (0.56) | (−2.37) | (2.23) |
| *sval* | −0.275*** | −0.141 | −0.108 | −0.237** | −0.155* | −0.033 | −0.260*** | −0.154* | −0.075 |
| | (−3.12) | (−1.63) | (−0.89) | (−2.54) | (−1.66) | (−0.31) | (−2.95) | (−1.78) | (−0.62) |
| *nyjg* | 0.066 | 0.065 | 0.011 | 0.049 | 0.136*** | −0.070 | 0.065 | 0.066 | 0.009 |
| | (1.44) | (1.45) | (0.17) | (1.02) | (2.83) | (−1.30) | (1.43) | (1.47) | (0.14) |
| *ict* | −0.576*** | −0.372*** | −0.200 | −0.608*** | −0.620*** | −0.060 | −0.567*** | −0.381*** | −0.179 |
| | (−5.72) | (−3.78) | (−1.45) | (−5.61) | (−5.71) | (−0.50) | (−5.63) | (−3.86) | (−1.31) |
| *govp* | −0.460*** | 0.091 | −0.601*** | −0.415*** | 0.075 | −0.413*** | −0.444*** | 0.078 | −0.567*** |
| | (−3.77) | (0.77) | (−3.61) | (−3.14) | (0.56) | (−2.80) | (−3.65) | (0.65) | (−3.43) |
| *hc* | | | | | | | 0.033* | −0.028* | 0.073*** |
| | | | | | | | (1.93) | (−1.67) | (3.10) |
| *_cons* | 9.695*** | 6.153*** | 4.677*** | 8.852*** | 9.397*** | 0.821 | 9.415*** | 6.391*** | 4.066*** |
| | (11.17) | (7.24) | (3.94) | (9.05) | (9.59) | (0.75) | (10.73) | (7.43) | (3.41) |
| Year FE | Yes | Yes | Yes | Yes | Yes | Yes | Yes | Yes | Yes |
| Province FE | Yes | Yes | Yes | Yes | Yes | Yes | Yes | Yes | Yes |
| N | 480 | 480 | 480 | 480 | 480 | 480 | 480 | 480 | 480 |
| F | 17.4893 | 10.7067 | 4.4783 | 15.7665 | 15.7669 | 2.884 | 16.2134 | 9.9574 | 5.0746 |
| R² | 0.820 | 0.710 | 0.733 | 0.820 | 0.821 | 0.750 | 0.822 | 0.712 | 0.739 |

Note:

*,

**, and

***indicate significance levels of 10%, 5%, and 1%, respectively; numbers in parentheses are standard errors.

these different models are consistent with the estimation results of the aforementioned benchmark model, which indicates that the conclusions of this study have good robustness.

## 7. Conclusion and implications

Based on the panel data of 30 provinces in mainland China, this paper applies the green economic growth accounting analysis to decompose and analyze the sources of green growth of the service industry, according to which the direction and potential of its development power transformation are explored. Meanwhile, based on existing research, an empirical analysis is carried out on the influencing factors of GTFP and its constituent factors in the service industry. The main conclusions of this part are as follows.

First, the level of green technological efficiency in the development of the service industry is low, and there is a downward trend, and the gap in green technological efficiency widens significantly across provinces. There is no necessary correlation between this technical inefficiency phenomenon and the level of service industry development. It is thus clear that there is still much room for the selection of appropriate technologies (including expanding the space for the selection of appropriate technologies by improving the absorptive capacity of technologies and selecting more appropriate production technologies under the premise of the existing absorptive capacity of technologies) and the optimization and adjustment of the scale of the service industry.

Second, GTFP in the service sector is generally declining, with large inter-provincial differences, and the obstacle to growth lies in the deterioration of green technical efficiency caused by the combined deterioration of green pure technical efficiency and green scale efficiency, and the growth driver lies in green technical progress.

Third, labor productivity in the service sector has been improved considerably, with large inter-provincial differences. However, the significant growth of the service sector has come at the cost of high inputs, and the growth contribution of GTFP and its constituent factors is negative, with the sector as a whole still in the input-factor-driven stage. However, the national benchmark provinces of Beijing and Shanghai have entered the innovation-driven stage of their service industries, and the growth contribution of GTFP is above 50%, even as high as 93.67% in Beijing. It is thus clear that the service industry urgently needs to transform its development drive from input-factor-driven to innovation-driven, and that there is much potential room for such a transformation.

Fourth, according to the empirical evidence from China, it has been found that the influencing factors of GTFP in the service industry are multifaceted, mainly involving structural factors, environmental regulatory factors, economic factors, social factors and institutional factors. Among them, there is a significant U-shaped relationship between the level of regional economic development and GTFP; the level of service industry development, the Internet penetration rate and the level of government financial expenditure have a significant inhibitory effect on GTFP in the service industry; while the structure of the service industry, the structure of the factor endowment, the structure of the energy consumption, and the intensity of the environmental regulation do not have a significant effect on GTFP, but basically, but basically all of them have a significant effect on some of these constituent factors.

This paper incorporates environmental constraints into the analysis of service industry transformation and upgrading, analyzes the power transformation process and future direction of service industry transformation and upgrading at the stage of high-quality development from the supply-side perspective with the help of green economic growth accounting and analysis model, and analyzes the influencing factors of transformation and upgrading of the innovation-driven service industry from the perspective of enhancing green total

factor productivity. Based on the above findings, this article proposes the following policy recommendations:

First, driving digital transformation in the service sector. Promote the construction of new infrastructure, strengthen the modularization of service elements and the professional division of labor, and improve the quality of services and the convenience of intelligent transformation and upgrading. Improvement of the digital service industry ecosystem and diversification of service modes and contents. Supply-side and demand-side joint efforts to promote industrial agglomeration from the original geospatial agglomeration mode to the network virtual agglomeration mode change.

Second, promoting value chain upgrading in the service sector. Introducing new technologies to refine and deepen the processing of traditional service industries. Fostering new service industries. Promoting competitive productive service industries. Industry-university-research cooperation and strengthening of talent cultivation. Optimize the modern service ecosystem.

Third, vigorously developing productive service industries. Enhancing the level of agglomeration of productive services and strengthening the degree of specialized agglomeration of productive services. Adopting information technology to respond to market demand in a timely manner. Select potential industries and enterprises to focus on breakthroughs.

## Supporting information

**S1 File. Appendix A** .
(DOCX)

**S1 Text. Minimal Anonymized Data** .
(DOCX)

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
