## [Decision Letter · Decision Letter 0]

23 Apr 2024

PONE-D-23-36022Power shift in the transformation and upgrading of the service sector --- Empirical evidence from ChinaPLOS ONE

Dear Dr. geng,

Thank you for submitting your manuscript to PLOS ONE. After careful consideration, we feel that it has merit but does not fully meet PLOS ONE’s publication criteria as it currently stands. Therefore, we invite you to submit a revised version of the manuscript that addresses the points raised during the review process.

We look forward to receiving your revised manuscript.

Kind regards,

Xingwei Li, Ph.D.

Academic Editor

PLOS ONE

Reviewers' comments:

Reviewer's Responses to Questions

**Comments to the Author**

1. Is the manuscript technically sound, and do the data support the conclusions?

Reviewer #1: Yes

Reviewer #2: Yes

2. Has the statistical analysis been performed appropriately and rigorously?

Reviewer #1: Yes

Reviewer #2: Yes

3. Have the authors made all data underlying the findings in their manuscript fully available?

Reviewer #1: Yes

Reviewer #2: No

4. Is the manuscript presented in an intelligible fashion and written in standard English?

Reviewer #1: No

Reviewer #2: Yes

5. Review Comments to the Author

Reviewer #1: Major comments:

(1) Overall, this paper lacks a clear theoretical contribution. That is, what can we learn from the Chinese case? The authors need to point it out clearly in the parts of introduction and conclusion.

(2) My first and primary concern lies in the novelty of this work, as I feel that the novelty issue has not been sufficiently highlighted in the current version. An important question shall be answered: does this work fill up some knowledge gaps which previous articles cannot address?

(3) I suggest the authors to rewrite the abstract with a focus on background,objectives methodology, main findings and conclusion. Please add a sentence which shows the necessity of the study.

(4) The"Conclusion" section should be largely improved, and the discussion and policy implication should be provided. The significance of the study should be enhanced in the Abstract,Introduction and Conclusion. The shortcoming of the study as well as more research and depth in the assessment and suggestions for the future should be presented.

(5) Whether the table in the appendix can be put into the Manuscript

Minor comments:

(1) Add a separator for the numbers over 1.000. Check all numbers including those in the tables/figures.

(2) The fonts, paragraphs and table formatting need further beautification.

(3) Remove the space after CO2 in line 302.

(4) The font should be centered within the table.

(5) From 2003-2019 should be From 2003 to 2019 in line 317.

(6) Line 446: Remove " and".

Reviewer #2: This article has done a lot of work, but there are some errors in some details. For example, there is a statement ambiguity in line 312; In line 357, decrease should be replaced by increase; The last sentence of line 908 is logically confused.

From the perspective of the explanation of the content of the article, line 329 lacks the necessary explanation for the occurrence of this situation; Line 609, the article uses the capital-labor ratio to measure the input of factors, and how to reflect the input of technical factors; On line 829, if the regression is not significant, is it still meaningful to discuss?

From the title of the paper, a large section and table in the later part of this paper discuss the influence of other variables on GTFP. The title focuses on the appropriateness of power shift, and the power shift in this paper is not clearly explained, and the specific decomposition process of development level and efficiency is not shown in detail, which needs further supplement and explanation.

Finally, after the conclusion of the analysis, the paper does not put forward specific and feasible suggestions, which needs to be supplemented and strengthened.

6. PLOS authors have the option to publish the peer review history of their article (what does this mean? ). If published, this will include your full peer review and any attached files.

**Do you want your identity to be public for this peer review?** For information about this choice, including consent withdrawal, please see our Privacy Policy .

Reviewer #1: **Yes: ** Xue Bing

Reviewer #2: No

---

## [Author Response · Author response to Decision Letter 1]

7 Aug 2024

Dear Editor and Reviewers:

We would like to express our heartfelt gratitude to two anonymous reviewers for their valuable feedback. Our manuscript, titled “Power shift in the transformation and upgrading of the service sector --- Empirical evidence from China”, benefited significantly from the constructive comments and insights from the review team. Based on the suggestions received, we have made careful revisions to the original manuscript. In the revised manuscript, all changes are marked in red. In addition, we have also carefully proofread this manuscript for typographical, grammatical, and other errors. We hope the revised manuscript is able to meet your standard of quality and address the concerns raised by the reviewers. In the following, you will find our detailed point-by-point responses to the reviewers’ comments.

---

## [Decision Letter · Decision Letter 1]

6 Jan 2025

Power shift in the transformation and upgrading of the service sector --- Empirical evidence from China

PONE-D-23-36022R1

Dear Dr. geng,

We’re pleased to inform you that your manuscript has been judged scientifically suitable for publication and will be formally accepted for publication once it meets all outstanding technical requirements.

Kind regards,

Xingwei Li, Ph.D.

Academic Editor

PLOS ONE

Additional Editor Comments (optional):

Reviewers' comments:

Reviewer's Responses to Questions

**Comments to the Author**

1. If the authors have adequately addressed your comments raised in a previous round of review and you feel that this manuscript is now acceptable for publication, you may indicate that here to bypass the “Comments to the Author” section, enter your conflict of interest statement in the “Confidential to Editor” section, and submit your "Accept" recommendation.

Reviewer #2: All comments have been addressed

2. Is the manuscript technically sound, and do the data support the conclusions?

Reviewer #2: Yes

3. Has the statistical analysis been performed appropriately and rigorously?

Reviewer #2: Yes

4. Have the authors made all data underlying the findings in their manuscript fully available?

Reviewer #2: Yes

5. Is the manuscript presented in an intelligible fashion and written in standard English?

Reviewer #2: Yes

6. Review Comments to the Author

Reviewer #2: Thank you for your revisions.

However, I suggest further refining the literature review. It would benefit from a more focused discussion of recent studies directly related to your topic, which will strengthen the context for your research.

7. PLOS authors have the option to publish the peer review history of their article (what does this mean? ). If published, this will include your full peer review and any attached files.

**Do you want your identity to be public for this peer review?** For information about this choice, including consent withdrawal, please see our Privacy Policy .

Reviewer #2: No

---

## [Editor Report · Acceptance letter]

PONE-D-23-36022R1

PLOS ONE

Dear Dr. Geng,

I'm pleased to inform you that your manuscript has been deemed suitable for publication in PLOS ONE. Congratulations! Your manuscript is now being handed over to our production team.

Kind regards,

on behalf of

Prof. Dr. Xingwei Li

Academic Editor

PLOS ONE